# Use of Remote Structural Tap Testing Devices Deployed via Ground Vehicle for Health Monitoring of Transportation Infrastructure

**DOI:** 10.3390/s22041458

**Published:** 2022-02-14

**Authors:** Roya Nasimi, Solomon Atcitty, Dominic Thompson, Joshua Murillo, Marlan Ball, John Stormont, Fernando Moreu

**Affiliations:** 1Department Civil, Construction, and Environmental Engineering, University of New Mexico, Albuquerque, NM 87131-0001, USA; rhnasimi@unm.edu (R.N.); jcstorm@unm.edu (J.S.); 2Department of Mechanical Engineering, University of New Mexico, Albuquerque, NM 87131-0001, USA; satcitty4@unm.edu (S.A.); domtom2000@unm.edu (D.T.); 3Department of Civil, Construction, and Environmental Engineering, Global and National Security Policy Institute, University of New Mexico, Albuquerque, NM 87131-0001, USA; jmurillo2@unm.edu; 4Department Electrical and Computer Engineering, University of New Mexico, Albuquerque, NM 87131-0001, USA; mball15@unm.edu

**Keywords:** tap testing, infrastructure, unmanned ground vehicle (UGV), robot, acoustic, data classification

## Abstract

Transportation infrastructure is an integral part of the world’s overall functionality; however, current transportation infrastructure has aged since it was first developed and implemented. Consequently, given its condition, preservation has become a main priority for transportation agencies. Billions of dollars annually are required to maintain the United States’ transportation system; however, with limited budgets the prioritization of maintenance and repairs is key. Structural Health Monitoring (SHM) methods can efficiently inform the prioritization of preservation efforts. This paper presents an acoustic monitoring SHM method, deemed tap testing, which is used to detect signs of deterioration in structural/mechanical surfaces through nondestructive means. This method is proposed as a tool to assist bridge inspectors, who already utilize a costly form of SHM methodology when conducting inspections in the field. Challenges arise when it comes to this method of testing, especially when SHM device deployment is done by hand, and when the results are based solely upon a given inspector’s abilities. This type of monitoring solution is also, in general, only available to experts, and is associated with special cases that justify their cost. With the creation of a low-cost, cyber–physical system that interrogates and classifies the mechanical health of given surfaces, we lower the cost of SHM, decrease the challenges faced when conducting such tests, and enable communities with a revolutionary solution that is adaptable to their needs. The authors of this paper created and tested a low-cost, interrogating robot that informs users of structural/mechanical defects. This research describes the further development, validation of, and experimentation with, a tap testing device that utilizes remote technology.

## 1. Introduction

The world’s transportation infrastructure continues to age. Infrastructure must be preserved to ensure the safety of our community members. The overall cost of maintenance and repair keeps increasing. The American Society of Civil Engineers’ (ASCE) Infrastructure Report Card evaluates the preservation of critical infrastructure and the 2021 grades scored low in every category. Roads were graded D, bridges C, transit D-, and aviation infrastructure D [1]. These types of infrastructure need to be modernized, even more so as the population of the United States has more than doubled since the 1960s, a time when many of the country’s major infrastructure systems were designed. All forms of infrastructure within the United States are being stretched thin [2]. The government transportation financial statistics (GTFS) provide information on transportation-related revenue and expenditures for all levels and modes of government transportation [3]. According to the GTFS, as of 2017, over USD 700 billion of transportation revenue was collected by the U.S. government. Nonetheless, the majority of this collected funding was allocated to maintenance of these systems and towards advancing infrastructure health monitoring technologies. Upwards of billions of dollars are being used to assess, maintain, and modernize the nation’s transportation infrastructure, yet this is the same infrastructure that is hardly meeting national health and safety standards [4]. Through the advancement and simplification of methods used for structural health quantification we can inform repair prioritization, simplify maintenance monitoring, and allow for reevaluation of government fund allocation.

Transportation infrastructure, especially highways and roads, are threatened by landslide and rockfall hazards. Each year, 25 to 50 people die in United States due to landslides [5]. Current mitigation methods require engineers in charge of the maintenance of transportation infrastructure to be able to identify signs of structural deterioration quickly and accurately. This information assists in the prompt repair or replacement of failing transportation systems. Non-contact devices for infrastructure inspections would be of interest if they could inform the classification of rock health automatically. Researchers use sensors and learning algorithms to classify the damage severity in structures [6,7]. This aids in the evaluation of structures in the field, including estimation of the severity of damage and deterioration [8]. Research is being conducted into the use of wireless smart sensing technology to measure bridge responses to inform bridge management decisions [9]. Wireless and remote sensors are modifiable to collect varied types of data, which are then stored and analyzed in real time to determine structural health status and aid in decision-making in the field. These data also aid in the development of effective methods to quantify structural conditions during inspections. 

Utilizing robotics and cyber-physical systems also benefits those conducting structural health tests. By combining human intelligence and robotic precision, one ensures proper repetition of tests, more accurate measurements, and the elimination of subjective assessments. Using remote-controlled robots to conduct tests in the field also serves to protect human lives and decrease the safety risks often involved in fieldwork. Transportation cyber–physical systems also increase efficiency and reliability by enabling increased feedback between cyber analysis systems and the physical transportation system [10]. By utilizing robotics and cyber–physical systems for structural maintenance and monitoring, there is a decrease in field testing safety challenges, and enables autonomy in assessing structures. This paper focuses on the automation of the classification of rock properties, informed only by a tap testing device that analyzes sounds collected in the field by a repeated tapping that can be automatically classified. The results of the paper show the success in the classification of the tests conducted on a rock surface near a highway using the tap testing method.

## 2. Motivation

The purpose of this research is to enable automatic classification of rock crack characteristics in the field. Accordingly, the study involved the design, development, and validation of a new low-cost, interrogating robot. The research team tested distributed surfaces with the machine to eliminate inspectors’ subjective judgements, and to automatically classify the two areas being hit. After testing the low-cost device’s performance in a laboratory environment, the required modifications for field deployments were applied. Subsequently, the authors conducted a field test using the modified tapping mechanism, BRUTUS II. During this research, the authors adapted a previously developed remote tap testing device that was deployable via an aerial robot [11], and describe its application for simpler scenarios. For analyzing the sound responses collected by the low-cost device, this research uses a linear classification method to identify rock surfaces with different crack characteristics. Although this project developed an integrated hardware–software system specifically for rockfall inspection, this system paves the road for non-contact inspections of other types of infrastructure. 

### 2.1. Hardware/Software

The hardware and software include the tapping mechanism, tapping hammer, remote-control transmission, and the data acquisition system. A remote-controlled ground vehicle, named BRUTUS, was developed to deploy the tap testing device. The acoustic data collected using the developed device were analyzed with an algorithm that combines Principal Component Analysis (PCA) with a linear classification method, k-means, as proposed by authors [12]. PCA uses the Singular Value Decomposition (SVD) concept to extract important information from multivariate data.

### 2.2. Tapping Mechanism

To recreate and simplify a tap testing device for use by a remotely operated vehicle, a four-bar linkage crank rocker concept was utilized (Figure 1). The purpose of this mechanism is to recreate the manual tapping motion of an inspector’s arm that occurs when a test is being conducted in the field. The crank rocker mechanism moves the tapping hammer’s head through a specified range of motion, enabling it to tap a given surface and, as a result, generates an acoustic response.

The crank rocker mechanism that was constructed consists of a gear box motor, a crank wheel, a rocker, a rocker arm, a coupler, and two position sensors. The tapping mechanism is driven by a 12 V gear box motor that is coupled with a larger crank wheel. The crank/motor mechanism is connected to a rocker arm via a coupler bar. This coupler bar translates the motion of the motor and crank wheel into the rocker arm. As the motor spins, the rocker arm moves forward and back through a specified range of motion. As the rocker arm moves, position sensors are used to track the total number of times the rocker mechanism has completed a cycle, as well as to mark a home position for the rocker arm once each cycle is completed. Position Sensor 1 tracks the total number of rotations that the motor has made since activation. Position Sensor 2 marks the retracted home position for the entire tapping mechanism. Once a preprogrammed and the desired number of cycles have been completed, the rocker arm mechanism returns to the home position. The right of the following image shows the crank rocker concept (Figure 1a), the completed build of the Planar Four-Bar Linkage.

### 2.3. Tapping Hammer

The tapping hammer solely comprises a 0.75-inch diameter Steel Ball Knob, connected to the end of the tapping mechanism’s rocker arm. When the crank rocker mechanism is activated, the Steel Ball Knob hits the surface of interest, and produces a specific acoustic response unique to the material state being tested. 

### 2.4. Remote-Control Transmission

The tap testing device was constructed, then mounted onto the frame of a Redcat Racing Gen8 International Scout II RC truck [13]. The Redcat utilizes an HX-1040-Crawler Hexfly Electronic Speed Control (ESC) to increase and decrease the amount of power that is supplied to the RC motor [14]. An FS-I6 X digital proportional radio system controls the RC. The transmitter is paired with an FS-iA6B 6 channel receiver [15]. The build concept for the entirety of the tap testing mechanism is illustrated in Figure 2.

The entirety of the BRUTUS 1 build is controlled by an Arduino Uno. This microcontroller is connected to the gear box motor through a HiLetgo Motor Driver Controller board [16]. The Arduino also specifies how many times the rocker arm mechanism must cycle before returning to the retracted home position, and communicates with the operator through the FS-I6 X transmitter and the FS-iA6B receiver. A piece of plywood was used to mount the tap testing device, among other materials, atop the Redcat chassis. The BRUTUS 1 tap testing device consists of a TSINY motor, a rocker arm mechanism (Four-Bar Rocker and Steel Ball Knob), a motor controller, two position sensors, a FS-I6 X transmitter, FS-iA6B receiver, an Arduino Uno, and Li-Po batteries. 

### 2.5. Data Acquisition System

Previous generations of the tap testing device used the TASCAM DR-44WL Linear PCM Recorder to collect acoustic data emitted by the tapping mechanism. The TASCAM’s DR-44WL has high-quality 2-channel stereo condenser microphones, with XY pattern and high-performance microphone components. Two locking XLR inputs allow for the lowest possible noise level and richest sound [17].

BRUTUS 1 intentionally used the same linear PCM recorder that was used in the aerial-vehicle-mounted version, enabling comparison with data from previous experiments. This helped to identify limitations, and improve data acquisition methods, and also assisted during the design of a new and improved tap testing device. The TASCAM DR-44WL was used to record acoustic response generated by the tapping mechanism’s hammer impacts. The two external microphones were installed near the base off the tapping mechanism on top of the chassis.

### 2.6. BRUTUS 1 as a Low-Cost Cyber Physical System

BRUTUS 1 took approximately 5 weeks to conceptualize, build, and test, and was produced at a cost just below USD 500 when accounting for all the parts used in the process. BRUTUS 1 consists of a TSINY motor, a rocker arm mechanism (Four-Bar Rocker and Steel Ball Knob), a motor controller, two position sensors, a FS-I6 X transmitter, FS-iA6B receiver, an Arduino Uno, a TASCAM PCM recorder, a Redcat International Scout RC truck chassis, an electronic speed controller, and Li-Po batteries. The following table (Table 1) summarizes the major building blocks of the BRUTUS 1 build, and the succeeding image (Figure 3) shows where each component is located on the BRUTUS 1 chassis (The numbered components in the figure correspond to listed components in Table 1).

## 3. Laboratory Validation of BRUTUS 1

An experiment was conducted to validate the performance and operation of this new generation of tap testing device. This experiment consisted of 12 tests on various stone specimens. There were 4 different stone specimen types that were each tested a total of three times; each of the specimens were cylindrical stones, all with varying defects. Each of the individual test specimens are shown in Figure 4: the first specimen was an unaltered stone (A); the second had a cracked top surface (B); and the last two were both split at shallow (C) and deep (D) depths and with respect to their front faces.

Prior to the beginning of the experiment, the programming parameters were preset to have the gear box motor run through its home position for a total of 100 revolutions upon receiving activation from the transmitter. Each test specimen was positioned high enough to ensure that the tap testing hammer would impact the center of the specimens (Figure 5). Once proper functionality of the device was confirmed, BRUTUS was piloted into a starting position in front of the test specimens. This position was used as the starting position for each test for the remainder of the experiment.

Upon activation, the tapping mechanism would proceed through 100 revolutions, with each cycle tapping the center of the test specimen. At the conclusion of the revolutions, the rocker mechanism returned to the neutral home position autonomously and the device was powered off. The sound data collected by the PCM recorder was then saved and exported to an external device. This process was repeated until each of the specimens had been tested three times total.

## 4. Laboratory Results

After creating the mechanism and hardware system, researchers tested the system in laboratory environment, before transferring it to field applications and conducting outdoor experiments. The team was able to successfully operate the device from a remote position and collect acoustic response data emitted by the test specimens. The data were collected and analyzed successfully in the laboratory environment [11]. Different discontinuities in the rock samples were classified using a method based on principal component analysis (PCA) and k-means clustering. The PCA method can analyze and classify sound signals using their variation. 

## 5. Field Validation and Deployment

To validate the performance of BRUTUS II for rock discontinuity identification, it was tested in an outdoor environment. A road located near a mountain with rockfall potential was selected. The authors took BRUTUS II to a roadcut in Tijeras, New Mexico, on August 16th, 2021. Figure 6 shows the selected site for field tests.

This site was of interest due to its high potential for rockfall hazards along the roadcut. This roadcut was composed of Madera limestone which was horizontally bedded with joint apertures that vary from 1 mm to 5 mm, and a joint roughness coefficient of 0 to 4. To equip the system for large-scale field experiments, the hammer of BRUTUS II was made with a larger one to impose higher impact energy and generate louder sounds from the rock surface. Additionally, it was modified to be water resistant for field operations. The proposed system does not measure input vibration as in traditional approaches; instead, the system analyzes sound waves reflected from the rock surface.

Five locations were selected along the roadcut wall (Figure 7). These locations were visually identified by geotechnical experts as being unstable or bearing cracks. BRUTUS II approached the selected locations and tapped their surfaces with the microphone switched on and collected the data.

Each location (1–5) was interrogated in sequence by piloting and positioning BRUTUS II adjacent to the test location to ensure proper strikes against the selected area.

BRUTUS II operated and collected the data successfully from the rock surface on a rainy day. The data collected at the roadside were noisier than the sounds recorded in the laboratory. To Evaluate the sound data recorded in the field, first, the time histories were plotted. Figure 8 displays the time history of the sound response from two separate locations collected in the field. The peaks in the time history represent one tap hit in the experiment.

These data were further analyzed using PCA and an algorithm proposed previously by the authors [12]. These analyses helped to identify and classify the tap sounds from different test locations. The row tap data were divided into two parts, training and testing sets. The taps from the training data were plotted in two-dimensional lower space. The area of the training data were trained into two regions, shown in different colors, and the testing taps were estimated using the trained machine for identifying different rock surfaces. Figure 9 shows the results of the analysis. The results show that the two areas being classified in the training phase (Figure 9a) match the two areas projected in the testing phase (Figure 9b). The results confirm that the field data supports the automatic classification, though assuming that there is enough past information from the representative areas to have a trained model. Another assumption is that the areas used for testing have similar characteristics to those in the training phase. The results of this approach are reasonable given the number of experiments being conducted and the purpose of field validation of rock classification. In the future, a larger number of sets (hundreds of thousands) and locations can compensate for assumptions about the homogeneity of the rock surfaces. This will be discussed in future work.

## 6. Discussion and Future Work

The laboratory and field tests prove that our overall device, data acquisition and classification methods are viable. By being able to collect and analyze these data, both in the controlled laboratory and field environments, BRUTUS I and II, respectively, demonstrated a simple yet effective method for conducting tap testing.

The objective of intended future work with the exhibited technology is to increase overall robustness and effectiveness. The primary goal is to ensure hammer head stability and security is ensured through adhesives and practical tests. Additionally, with BRUTUS, another goal is to make the system modular and easier to transport from platform to platform. To accomplish this, it is necessary to make a more compact and versatile rocker mechanism alongside a more streamlined circuit. Preliminary experiments and designs have already been completed, in which the updated system was mounted upon, and utilized with, an Unmanned Surface Vehicle (USV) (Figure 10). Additional upgrades to improve the effectiveness of BRUTUS include software modification to optimize the data acquisition system, improvements in the device’s remote controllability, and use of nonlinear classification methods.

Furthermore, with the continued development of sensor technology, and the increased access to 3D printing hardware and software, the authors aim to continue developing and revolutionizing methods used for structural monitoring. Improvements include compacting the system and lowering its cost, in addition to the aforementioned aspirations. By doing so, this provides one with the ability to inspect parking structures, historic sites, old masonry buildings, or other critical structures with automatic tap testing within city environments and elsewhere. The possibilities for expansion are extensive; however, the goal is to empower others to develop and adapt this technology for their own needs. By successfully creating a low-cost cyber–physical system that pinpoints and interrogates surfaces, BRUTUS ultimately simplifies the tap testing process, which allows for expanded use by those who wish to use the technology.

## 7. Conclusions

The proposed tap testing mechanism has been tested on real rock surfaces of interest to the Department of Transportation. The previous generation of remote tap testing devices were meant for solely testing concrete and lumber; however, the proposed method takes a step forward and uses this technology for various geotechnical applications as well. This paper introduces an automated tapping mechanism that works with a linear classification algorithm to identify different discontinuities on rock surfaces in the field. The system is designed using a bottom–up approach to be low-cost, user-friendly, and robust enough for field applications. The team tested the first version of the device in the laboratory, and after validating the performance of both the hardware and software, the design was improved and modified for field tests. This device gives inspectors, as well as community members, an advantage when it comes to surveying and identifying problems within transportation infrastructure. The design of this simple, low-cost, interrogating robot device, coupled with a PCA-based classification method, should inspire and incentivize the development of other low-cost sensors. Several tests in the laboratory and field show that the proposed system can operate, collect data, and use data to obtain information in the field. This device has the potential to be valuable to the stakeholders, while demonstrating how technology is beneficial for protecting and maintaining infrastructure. Individuals who are willing and able to recreate, and in some cases improve, the existing technology, can decrease the cost of structural health monitoring as the technology is proliferated. By designing, developing, and validating new low-cost smart sensing robot technology that is easy to advance and deploy, devices such as BRUTUS help to decrease inspection complexities, ensure quick repair of failing systems, and give power back to the community.

## Figures and Tables

**Figure 1 sensors-22-01458-f001:**
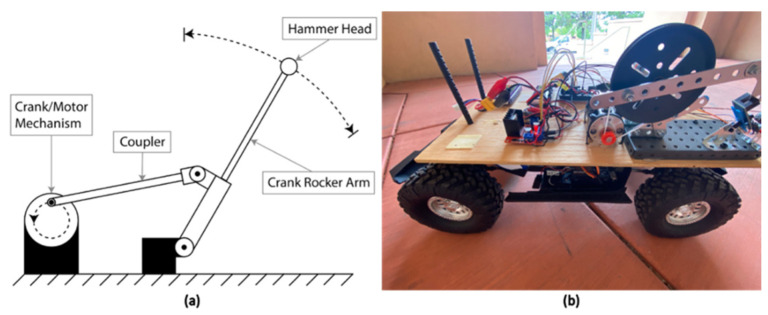
Tapping mechanism (**a**) Side view of Four-Bar Linkage Crank Rocker concept; (**b**) Side view of constructed Four-Bar Linkage Crank Rocker.

**Figure 2 sensors-22-01458-f002:**
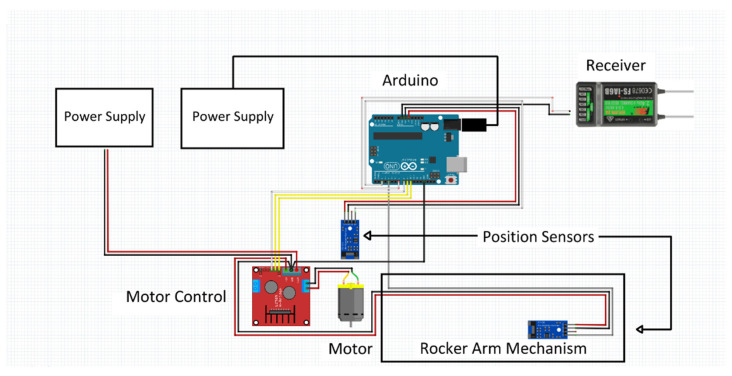
Build concept for tap testing mechanism.

**Figure 3 sensors-22-01458-f003:**
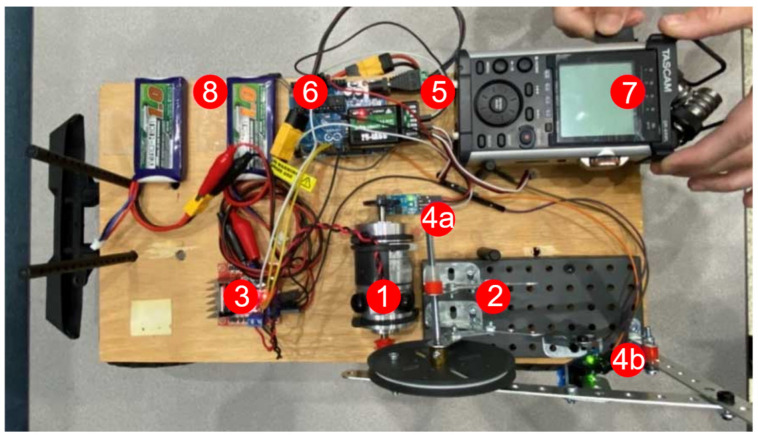
BRUTUS 1 component diagram, components labeled with numbers.

**Figure 4 sensors-22-01458-f004:**
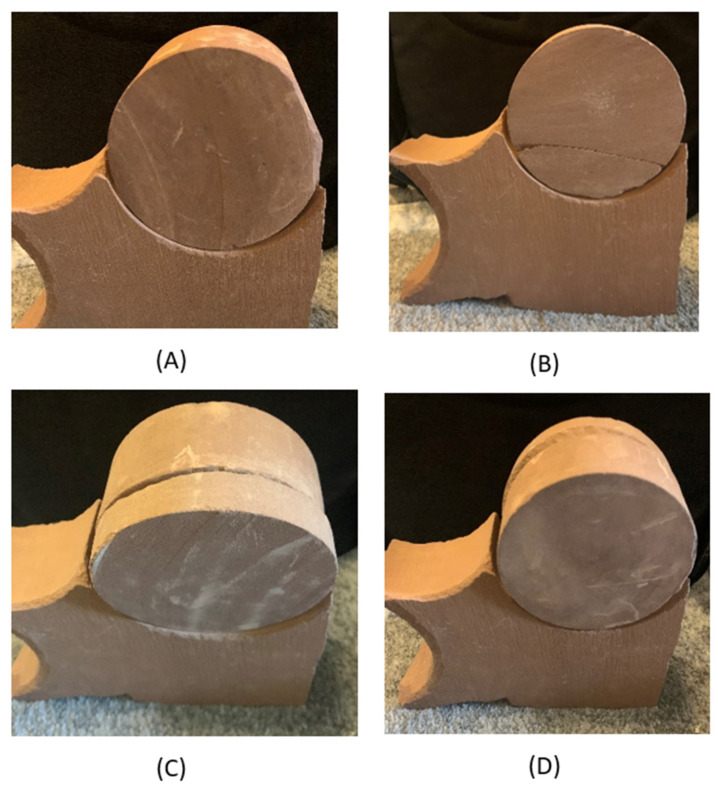
Stone specimens used in testing of BRUTUS 1: (**A**) unaltered stone; (**B**) stone with crack on top surface; (**C**) stone with shallow split; (**D**) stone with deep split.

**Figure 5 sensors-22-01458-f005:**
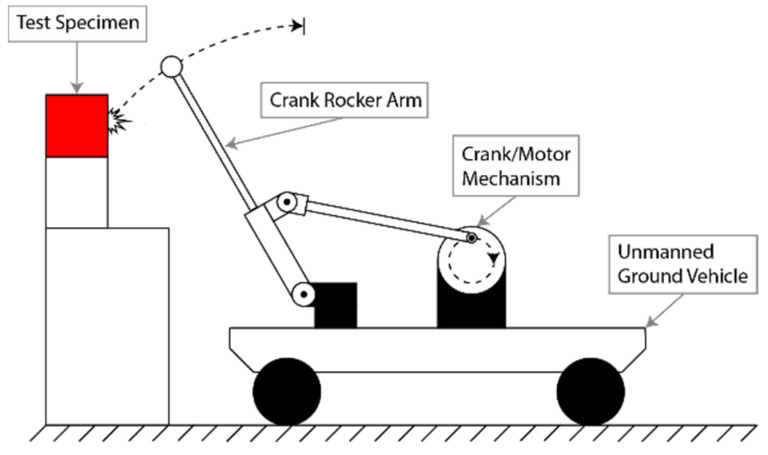
Experimental setup and concept drawing.

**Figure 6 sensors-22-01458-f006:**
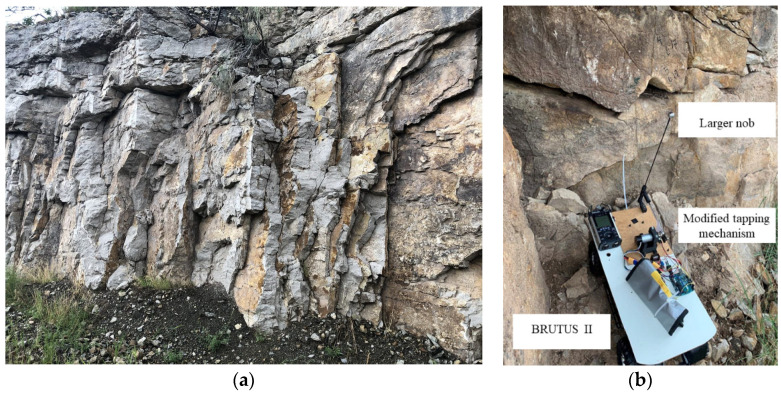
Field experiment: (**a**) Roadcut testing site; (**b**) modified UGV, BRUTUS II.

**Figure 7 sensors-22-01458-f007:**
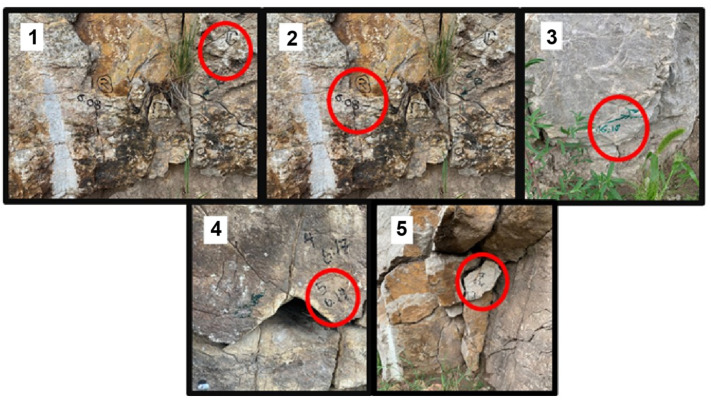
Test locations.

**Figure 8 sensors-22-01458-f008:**
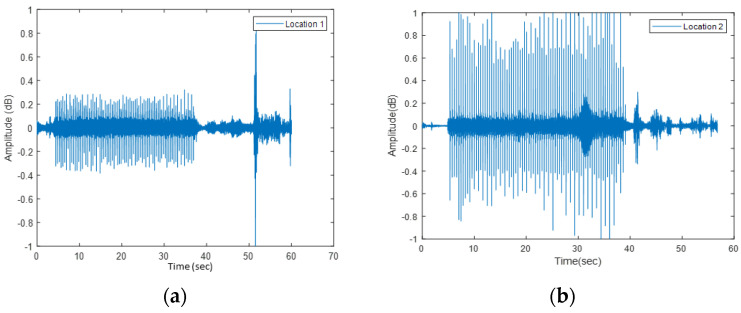
Time history of taps on: (**a**) Location 1; (**b**) Location 2.

**Figure 9 sensors-22-01458-f009:**
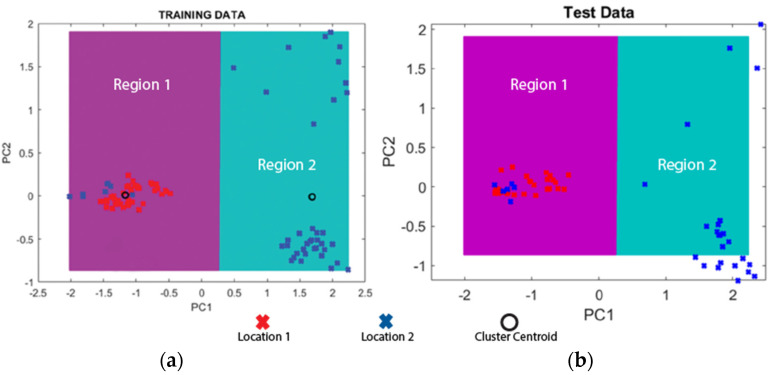
Data classification of location 1 and location 2: (**a**) Training data; (**b**) Test data.

**Figure 10 sensors-22-01458-f010:**
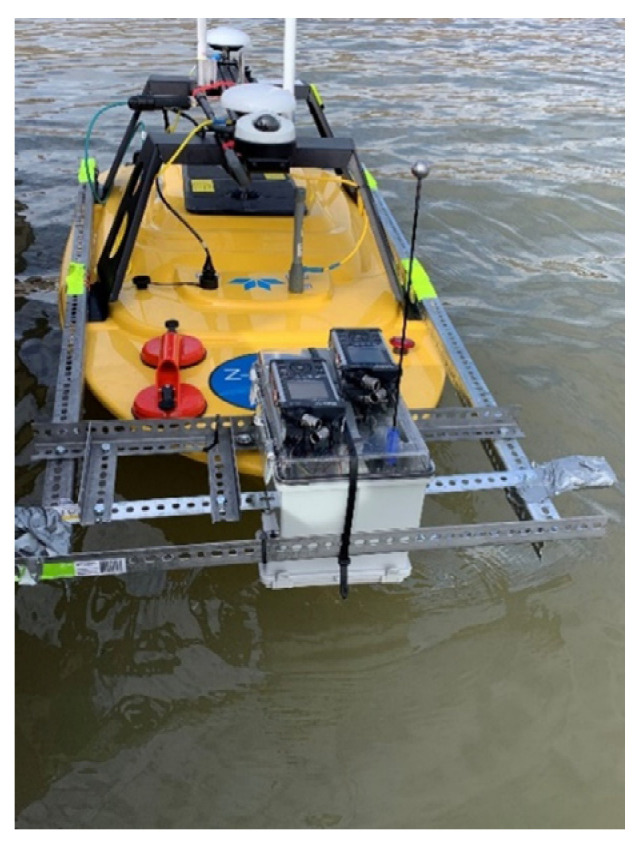
Newer BRUTUS system on USV.

**Table 1 sensors-22-01458-t001:** Major components of BRUTUS 1.

Component	Purpose
TSINY Motor (1)	Drives the rocker arm mechanism.
Rocker Arm Mechanism (Four-Bar Rocker and Steel Ball Knob) (2)	Drives tap testing hammer head; hammer head creates acoustic response (Steel Ball Knob).
Motor Controller (3)	Gives full control of motor speed and function through communication via the Arduino Uno.
Position Sensors (Arduino Comp.) (4a/b)	4a—Tracks completed tapping cycles; 4b—marks retracted home position for rocker arm mechanism.
Transmitter (5)	Used for remote control of the BRUTUS 1 device; communicates with device through the FS-iA6B receiver.
Receiver (6)	Relays information between FS-iA6B transmitter
Arduino Uno (7)	An input–output device used to help control other components of BRUTUS 1 by utilizing personally developed code.
PCM Recorder (8)	Record and stores acoustic response data.
RC Truck Chassis (Not Shown)	Base RC chassis for BRUTUS 1; enables remote movement of device when paired with an ESC.
ESC (Not Shown)	Used to control speed of RC truck chassis motors; enables manual control of RC vehicle’s movement/turning speed.
Li-Po Batteries (9)	Used to power BRUTUS 1.

## Data Availability

Data available on request due to confidentiality with the stakeholder collaborator. The data presented in this study are available on request from the corresponding author. The data are not publicly available due to the confidential nature of the experiment location and the stakeholder collaborator.

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
