# Peer review of "Use of Remote Structural Tap Testing Devices Deployed via Ground Vehicle for Health Monitoring of Transportation Infrastructure"

_sensors, 2022, doi:10.3390/s22041458_

Round 1

Reviewer 1 Report

At the beginning, I must admit that I have some difficulty in properly assessing the presented article. It is possible that I do not know the art of authors. It is equally likely that this is an engineering degree job. I am inclined to the second option.

Moving on to the details:

  1. A very well written and convincing introduction. It suggests that serious scientific work will follow. The chapter on motivation is also convincing.
  2. The content, i.e. the design of the vehicle, the mechanism enabling the impulse test implementation is at the high school level. I understand the concept of low cost, emphasized many times, and the "initial" stage of work. However, this does not change my assessment.
  3. The drawings are of very poor quality and don't really explain anything. Both, those depicting the construction and others. They are doubled in total, I guess in order to fill the volume of the article.
  4. The work does not present any research results, nor a new design solution or algorithm;
  5. As for the functionality of the vehicle itself, no reference was made to the need to ensure adequate impact energy. Exciting building structures to vibrate requires a significant amount of energy. Tests on blocks with different damage distribution do not explain or validate anything. It was just that the "car" drove up to the pulley and tapped it. Nothing comes of this.
  6. The chapter on the results is redundant, as there are no results in fact.
  7. Chapters 4 and 5 are so general that they could be read without reading the previous chapters.

To sum up: An article at a very low level, showing no results. Ordinary work at the level of a high school diploma.

Author Response

At the beginning, I must admit that I have some difficulty in properly assessing the presented article. It is possible that I do not know the art of authors. It is equally likely that this is an engineering degree job. I am inclined to the second option.

{Response}: The authors thank the reviewers for their specific and very constructive feedback, and we agree that their comments are beneficial to improve the contribution of our work to Sensors’ readers. We have addressed all the comments one by one. The authors made 55 revisions in addition to 686 insertion and deletion while keeping the scope of the manuscript. We believe the updated manuscript is ready for publication. However, we are available for further clarification as needed.

Moving on to the details:

  1. A very well written and convincing introduction. It suggests that serious scientific work will follow. The chapter on motivation is also convincing.

{1-1 Response}:

Discussion: The authors thank the reviewer for the comment.

  1. The content, i.e. the design of the vehicle, the mechanism enabling the impulse test implementation is at the high school level. I understand the concept of low cost, emphasized many times, and the "initial" stage of work. However, this does not change my assessment.

{1-2 Response}:

Discussion: The authors thank the reviewer for the comment. We benefitted reviewer’s comment to improve the paper. Following their recommendation we decided to emphasize on the other aspects of the system. Contribution of the paper is the system that helps managers to collect sound data in the field, change decisions about the rocks. We have added new chapters that are more value for road; rock; and transportation maintenance.

Changes in manuscript:

Chapters 5 and 6 added.

  1. The drawings are of very poor quality and don't really explain anything. Both, those depicting the construction and others. They are doubled in total, I guess in order to fill the volume of the article.

{1-3 Response}:

Discussion: Thank you very much for your comment. We revised all figures and drawing and the way that they are presented in the paper.

Changes in manuscript:

On page 3 of manuscript:

We replaced Drawing in Figure 1 with better one. Also, Figure 1 and Figure 2 are combined to remove the redundancy.

On page 7 of manuscript:

We removed former Figure 3 in last version of the manuscript.

On page 5 of manuscript:

Figure 3 is edited and rotated to save the space.

On page 7 of manuscript:

Figure 5 is edited and has a drawing accompanying the photograph.

  1. The work does not present any research results, nor a new design solution or algorithm;

{1-4 Response}:

Discussion: The authors thank the reviewer for their comment; we agree with the reviewer. We added a new section that presents field results. We have published preliminary laboratory results in the past and it was not in the field but the result in the new version shows the results in the field. Authors believe new additions increase the value of the paper. Thanks!

Changes in manuscript:

On page 9 of manuscript:

Sections 5 and 6 are added.

On page 8 of manuscript:

This plot illustrates the PCM recorder’s ability to properly collect acoustic data. Later, by processing this data, it becomes useful in determining whether tested material and surfaces are damaged, lightly damaged, or undamaged. This is done by utilizing the Principal Component Analysis (PCA) concept to classify the acoustic data into different categories [11].

  1. As for the functionality of the vehicle itself, no reference was made to the need to ensure adequate impact energy. Exciting building structures to vibrate requires a significant amount of energy. Tests on blocks with different damage distribution do not explain or validate anything. It was just that the "car" drove up to the pulley and tapped it. Nothing comes of this.

{1-5 Response}:

Discussion: Thanks for the constructive feedback. We agree that new additions will increase the quality of the paper. We added a sentence to clarify to method is not traditional input vibration we collect sound response from hits. Additionally, we added the results of driving car.

Changes in manuscript:

On page 9 of manuscript:

  • The system proposed is not traditions input vibration. The system analysis a sound reflected from surface.
  • We added results from driving car.

On page 12 of manuscript:

  • For the system’s compatibility with large scale field experiments the hammer of BRUTUS was replaced with a larger one to impose higher impact energy and record louder sound from the surface.

  1. The chapter on the results is redundant, as there are no results in fact.

{1-6 Response}:

Discussion: Thanks for the comment. Authors addressed this comment with adding new sections, results and removing redundancy as detailed in previous comments. We added new results from field experiments in the new version of the paper. Authors did an overall review of the paper to remove redundancy.

  1. Chapters 4 and 5 are so general that they could be read without reading the previous chapters.

{1-7 Response}:

Discussion: The authors added new sections and elaborated and improved the manuscript in the over-mentioned sections. Thanks!

Changes in manuscript:

On page 9 of manuscript:

  • Sections 5 and Section 6 are added.
  • Last chapters are reorganized and revised.

To sum up: An article at a very low level, showing no results. Ordinary work at the level of a high school diploma.

{Response} The authors agree that the former version of the paper did not show results, conclusions were lacking substance, and the former emphasis was too narrowed in the hardware and software of the innovation. We believe the newer version addresses the important feedback provided by this reviewer, and we have significantly updated the paper with this constructive direction. We believe the new version of the paper has improved a lot by your remarks, and we think the readers now will benefit from this newer version.

Reviewer 2 Report

This research provides a robotic-based procedure to identify the structural damage exposed to infrastructures.  The research topic is interesting and within the scope of the journal. However,  the authors should address the following major comments in their manuscript. 

  1. The writing style of the manuscript is not suitable for publication. For example,  there are several long and wordy sentences,  which should be improved by the authors.
  2. It is more appropriate if the authors provide a validation test  to present the error related to data acquisition.   
  3. How can the authors prove the robustness of the device in detecting the structural damage using the proposed method. 
  4. The introduction section should be expanded to include more relevant references,  such as Vibration anatomy and damage detection in power transmission towers with limited sensors,  and A decision tree-based algorithm for identifying the extent of structural damage in braced-framr buildings. 

Author Response

General Response from the Authors

The authors thank the reviewers for their specific and very constructive feedback, and we agree that their comments are beneficial to improve the contribution of our work to Sensors readers. We have addressed all of the comments one by one. The authors made 55 revisions in addition to 686 insertion and deletion while keeping the scope of the manuscript. We believe the updated manuscript is ready for publication. However, we are available for further clarification as needed.

This research provides a robotic-based procedure to identify the structural damage exposed to infrastructures.  The research topic is interesting and within the scope of the journal. However, the authors should address the following major comments in their manuscript. 

  1. The writing style of the manuscript is not suitable for publication. For example, there are several long and wordy sentences, which should be improved by the authors.

{2-1 Response}:

Discussion: Thank you for your response the authors agree with reviewer’s comment. The whole manuscript was revised in writing style and grammar.

Changes in manuscript:

The authors made 55 revisions in addition to 686 insertion and deletion while keeping the scope of the manuscript.

  1. It is more appropriate if the authors provide a validation test to present the error related to data acquisition.   

{2-2 Response}:

Discussion: The authors thank the reviewer for their suggestions we added sections for validation tests that were conducted in the field.   

Changes in manuscript:

On page 9 of manuscript:

Sections 5 and 6 are added.

On page 8 of manuscript:

This plot illustrates the PCM recorder’s ability to properly collect acoustic data. Later, by processing this data, it becomes useful in determining whether tested material and surfaces are damaged, lightly damaged, or undamaged. This is done by utilizing the Principal Component Analysis (PCA) concept to classify the acoustic data into different categories [11].

  1. How can the authors prove the robustness of the device in detecting the structural damage using the proposed method. 

{2-3 Response}:

Discussion: Thank you for your suggestion. We did multiple experiments in laboratory and outdoor. We added contents that proves the robustness of the proposed system and the method. We believe it improved the paper’s quality. Newer version of the manuscript better presents the value and application of the system working with method.

Changes in manuscript:

      On page 9 of manuscript:

  • Sections 5 and 6 are added.

      On page 10 of manuscript:

  • Section 6 was elaborated and discusses the robustness and potential of the system beyond the scope of the car vehicle.

  1. The introduction section should be expanded to include more relevant references,  such as Vibration anatomy and damage detection in power transmission towers with limited sensors,  and A decision tree-based algorithm for identifying the extent of structural damage in braced-framr buildings. 

{2-4 Response}:

Discussion: Thank you for your suggestion. Authors improved the introduction and added references to the manuscript.

Changes in manuscript:

On page 2 of manuscript:

“Researchers use sensors and learning algorithms to classify the damage severity in structures [5, 6].”

Round 2

Reviewer 1 Report

Despite the introduction of a large number of changes, the improvement of the quality of the illustrations and the additions made, the article presents a low scientific value.

It is a description of a technical solution of a remotely controlled vehicle that is capable of generating a hammer hit to a rock. The response is recorded by the microphone. So much and only that.

It is not entirely clear what was the purpose of the research - the crack detection algorithm or the demonstration of the maneuverability of the vehicle?

Regarding selected issues:

  1. The introduction skilfully outlines the problem of the constant aging of road / construction infrastructure and the need to monitor its condition. However, in Chapter 5, the validation takes place in the context of the danger of falling rocks. This is a direct reference to the research described in the position cited as [11] by the same authors. Even the name "Brutus" was added in accordance with what was contained in the publication [11], which in my opinion is much more interesting than the currently reviewed one. Even the same photos were used in both publications.
  2. It is a pity that photos from the implementation of the validation experiment have not been presented. It would be interesting to evaluate how this stage was carried out. It is hard for me to imagine how "Brutus" could excite the rock in Figure 7 to vibrate.
  3. I do not know the reason for including Figure 6 in the article. It is not explained, because it is probably not only about the visualization of 74 hits (instead of the assumed 100). You can't see anything between the peaks. There is no reference to this figure in the text of the paragraph (number 6 in line 250 is missing). The same applies to Figure 9.
  4. In my opinion, the conclusions are not strongly supported by the text of the article and are of a very general nature.

Author Response

The authors thank the reviewer for their time and feedback. We agree that their comments are beneficial to improve the contribution of our work to Sensors readers. We have addressed all of the comments one by one and believe the newer version of the manuscript is good for publication but the authors would like to address other issues that reviewers may find needed.

Despite the introduction of a large number of changes, the improvement of the quality of the illustrations and the additions made, the article presents a low scientific value.

{1-1 Response}:

Discussion:  We added more explanations in motivation section of the revised manuscript to provide the scientific value of the article in the terms of design and development of a new low-cost device working with a novel machine learning method. We agree the machine learning component in the field was not clearly outlined earlier. The new additions emphasize the successful low-cost machine learning implementation and classification result from the proposed method in large scale and outside of the controlled laboratory.

Changes in manuscript:

On page 2 of manuscript:

In this paper, first the development of a low-cost robot called BRUTUS is explained. After testing the low-cost device’s performance in laboratory environment, the required modifications for field deployments were applied. Subsequently, the authors conducted a field test using modified tapping mechanism, BRUTUS II.

It is a description of a technical solution of a remotely controlled vehicle that is capable of generating a hammer hit to a rock. The response is recorded by the microphone. So much and only that.

{1-2 Response}:

Discussion: The authors thank the reviewer for bringing up their concern. We acknowledge their comment in the first revision and added explanation about the methodology and classification results in the field. However, we agree that the emphasis needs to be more clear, and added an additional sentence to further clarify the wider scope of this work.

Changes in manuscript:

On page 3 of manuscript:

Even though this project developed the integrated hardware-software system for rockfall inspections, this system paves the road for non-contact inspections for other types of infrastructure.

It is not entirely clear what was the purpose of the research - the crack detection algorithm or the demonstration of the maneuverability of the vehicle?

{1-2 Response}:

Discussion: Thank you for the question! Actually, the purpose of the research was both. The purpose was an integrated low-cost software and hardware system that could help inspectors to conduct safer operations. However, the main emphasize in this manuscript is put on the hardware development and improvement for field operations. In this project the hardware and software were developed to work together. Brutus and method were first developed for a simple small scale laboratory test. However, the system needed modifications before transferring to field size experiments. Brutus was upgraded and it was still not clear that the method would work with the noise of roadside vehicles in the field test. For implementation the team made the sensors on the Brutus waterproofed and improved the tapping mechanism which are indicated in the revised version of the paper.

Changes in manuscript:

On page 2 of manuscript:

In this paper, first the development of a low-cost robot called BRUTUS is explained. After testing the low-cost device’s performance in laboratory environment, the required modifications for field deployments were applied. Subsequently, the authors conducted a field test using modified tapping mechanism, BRUTUS II.

On page 12 of manuscript:

The system is designed using a bottom-up approach to both be low-cost and us-er-friendly, and robust enough for field applications. The team tested the first version of the device in laboratory and after validating the performance of both hardware and software the designed was improved and modified for field tests.

Regarding selected issues:

  1. The introduction skillfully outlines the problem of the constant aging of road / construction infrastructure and the need to monitor its condition. However, in Chapter 5, the validation takes place in the context of the danger of falling rocks.

{1-4 Response}:

Discussion: The authors thank the reviewer their insightful comment. We agree with the reviewer we added the justification for the experiments we did for rock falls. The new addition connects the two parts of the paper rationally. We appreciate the reviewer’s catch on an important point.

Changes in manuscript:

On page 2 of manuscript:

Transportation infrastructure especially highways and roads are threatened by landslide and rockfall hazards. Each year 25 to 50 people die in United States due to landslides [5]; that’s why non-contact low-cost devices for infrastructure inspections are preferred.   

This is a direct reference to the research described in the position cited as [11] by the same authors. Even the name "Brutus" was added in accordance with what was contained in the publication [11], which in my opinion is much more interesting than the currently reviewed one. Even the same photos were used in both publications.

{1-4b Response}:

Discussion: Thank you for your careful review. We removed figure 5b from the article that was repetitive. In addition, the propose of the chapter 3 of the manuscript is demonstrating that the first version of the vehicle gives reasonable results before moving to the field. However, we agree that elaborated explanation of the section can divert the main message of the paper. We cited the previous work because the objective of the section is to demonstrate the successful operation of the Brutus then transfer it for data analysis in the field avoiding repetition. In consideration of reviewers comment we removed some contents.

Changes in manuscript:

On page 7 of manuscript:

Figure 5(b) is removed.

On page 8 of manuscript:

Irrelevant information at the end of the chapter 4 is removed.

  1. It is a pity that photos from the implementation of the validation experiment have not been presented. It would be interesting to evaluate how this stage was carried out. It is hard for me to imagine how "Brutus" could excite the rock in Figure 7 to vibrate.

        {1-5 Response}:

Discussion: The authors thank the reviewer for the comment. We added a figure from the field implementation using Brutus II. Additionally, the authors would like to emphasize that the proposed method is not a vibration-based approach but an acoustic response-based method. On a side note: authors distinguish between two different UGV systems developed in this research using Brutus II for the upgraded version.  

Changes in manuscript:

On page 8 of manuscript:

Figure 6(b) added from experiment.

Clarifications for the system and the way it works:

  • Additionally, it was designed to be water resistant for field operations
  • BRUTUS II operated and collected the data successfully from the rock surface in a rainy day. The data collected at the roadside are noisier than the sound recorded in laboratory.
  • For analyzing the sound response collected by the low-cost device

  1. I do not know the reason for including Figure 6 in the article. It is not explained, because it is probably not only about the visualization of 74 hits (instead of the assumed 100). You can't see anything between the peaks. There is no reference to this figure in the text of the paragraph (number 6 in line 250 is missing). The same applies to Figure 9.

       {1-6 Response}:

Discussion: Thanks for the comment. Following the reviewer’s 1-4b comment about ambiguousness of the topic of interest we removed Figure 6. Also, the authors added more explanation for Figure 9.

Changes in manuscript:

On chapter 4 of manuscript:

Figure 6 is removed.

On page 9 of manuscript:

For evaluation of the selected locations, the data gathered on these areas were first examined. Figure 8 displays the time history of the sound response from two separate locations collected in the field. The peaks in the time history represent one tap hit in the experiment.

  1. In my opinion, the conclusions are not strongly supported by the text of the article and are of a very general nature.

Discussion: Thank you for your recommendation. We agree with the reviewer and have revised the conclusions. Thank you for the opportunity to allow us to improve the contribution of the paper.   

Changes in manuscript:

On page 11 of manuscript:

The system is designed using a bottom-up approach to both be low-cost and us-er-friendly, and robust enough for field applications. The team tested the first version of the device in laboratory and after validating the performance of both hardware and software the designed was improved and modified for field tests.

Reviewer 2 Report

i recommend this paper for publication

Author Response

Thank you for your insights and constructive reviews